# The Paradoxes of Modern Islamic Discourses and Socio-Religious Transformation in the Digital Age

**Sahar Khamis**

Department of Communication, University of Maryland, College Park, MD 20742, USA; skhamis@umd.edu

**Abstract:** The introduction of the internet brought about many transformations in the political, social, cultural, and educational fields worldwide. This phenomenon of digital transformation introduced a myriad of positive, negative, and paradoxical impacts. This critical essay tackles some of the significant transformations and paradoxes which the introduction of the internet invited in modern Muslim societies, with a special focus on two specific domains. First, the realm of religious authority or obtaining authoritative religious knowledge in the age of the internet. Second, the realm of shifting gendered Islamic identities in the age of cyberspace. In exploring these complex and hybrid phenomena, special attention is paid to the tensions between the opposing forces of tradition and modernity, diversity and cohesion, hegemony and resistance, and globalization and localization in cyberspace, and their numerous and far-reaching effects.

**Keywords:** internet; digital age; cyberspace; Islamic authority; Islamic feminism; gender

## 1. Introduction: Digital Transformations in the Muslim World

The introduction of the internet, and its many related applications, created the capacity for new forms of identity formation and transformation at the individual, communal, societal, and global domains in many ways which were not possible in the past. This created far-reaching impacts on modern Muslim communities in the age of cyberspace (Bunt 2000, 2003; Eickelman and Anderson 2003; Mandaville 1999, 2001, 2003, 2007), including the creation of digitalized transnational Muslim communities (Bunt 2000, 2003; Cesari 2004), Muslim "transnational public spheres'" (Eickelman and Anderson 2003), and an imagined global *umma* (Muslim community) in the digital age (El-Nawawy and Khamis 2009).

These digitally enabled emerging phenomena had especially impactful effects on the re-shaping of young people's thought patterns, beliefs, lifestyles, and practices (Karim 2003), taking into account that youth are the most avid and frequent users of the internet and its many related platforms. Muslim youth are no exception. For new generations of Muslim youth, especially those who are living in the diaspora outside of their homelands, internet-enabled applications provided important windows, bridges, and platforms for them to adjust to their new communities, on one hand, while maintaining much-needed connections with their home countries, on the other hand (El-Nawawy and Khamis 2009).

In so doing, I argue that Muslim youth discovered vital new tools to challenge pre-existing notions of Islamic identity and what it means to be a Muslim in the modern age, through re-thinking a lot of the taken-for-granted thought patterns and practices. Ideally, this replaces a "passive audience recipient with a potentially active-surfing public" (Dartnell 2005) who are actively engaging with innovative forms of communication while producing equally innovative ideas, through acting as producers and consumers of digital media content simultaneously.

It can be argued that the internet plays a vital role in satisfying the inquisitive desire of young Muslims to learn more about their own culture and faith. They surf the net to connect with each other and to obtain information on how to cope with various cultural

encounters without deviating from their faith, in addition to learning how to set a good example for Islam in the Western world (Mandaville 2001, 2003; El-Nawawy and Khamis 2009). This provides evidence of the transformative power of the internet in terms of cross-cutting the boundaries between local and global spheres, as well as traditional and modern domains in the modern age.

In acknowledging all of these unprecedented opportunities which were made possible for modern Muslims through the availability of the internet and its many roles and functions, it is equally important to shed light on some of the paradoxes which resulted from these new phenomena in the age of cyberspace.

Therefore, the rest of this critical essay tackles two domains where the paradoxical impacts of the internet and its many associated applications became particularly visible and prevalent, namely shifting Islamic religious authority and shifting gendered Muslim identities.

## 2. The Paradoxes of Islamic Authority in the Digital Age

Despite the many opportunities and advantages which were made possible in the age of cyberspace, a number of dangers and threats are also worth noting. One of them is the curtain of anonymity behind which many internet users hide. This makes it increasingly difficult to tell the difference between a trustworthy, credible, and knowledgeable Islamic scholar, or *'alim*, and a layperson or an amateur who is posing as one (Mandaville 2007). This changes the boundaries of Islamic religious knowledge, while inviting the threat of a cacophony of voices in cyberspace in the realm of sending, sharing, and receiving such knowledge. This becomes especially dangerous when some of the unqualified or underqualified internet users give themselves the liberty of issuing *fatwa*, or religious advice (El-Nawawy and Khamis 2009).

This dilemma of "who speaks for Islam" becomes especially threatening when coupled with the spread of religious misinformation and disinformation in the age of cyberspace, due to the immediacy, wide outreach, and global scope through which internet-based content and internet-enabled communication could spread and proliferate.

These threats are magnified by the fact that many Islamic religious scholars, or *ulama*, who master various forms of Islamic religious knowledge, do not equally master digital literacy skills, communication skills, or linguistic skills. This means that they have been traditionally excluded from digital spaces, which are the preferred domains for modern-day young and technologically savvy Muslim communities, especially those who are living abroad in the diaspora (El-Nawawy and Khamis 2009).

It is worth mentioning, however, that this pattern has been gradually shifting in recent years with the emergence of a new generation of young Islamic scholars who combine religious knowledge with modern day communication skills, technological savviness, and linguistic and cultural competency, making them especially equipped to deal with Muslim youth both at home and in the diaspora and to provide them with much-needed religious knowledge online.

One good example is Amr Khaled, an Egyptian accountant turned preacher and a young modern religious figure, who was not formally or traditionally trained in Al-Azhar or other credible institutions of Islamic education. Yet, due to his novel approach, his informal and personable style, his special appeal among young people, and his successful reliance on new media technologies, he was able to secure high visibility, sound credibility, and a large base of young loyal fans online (El-Nawawy and Khamis 2009).

Other examples have emerged recently, such as Moustafa Hosni, a young Egyptian non-traditional preacher who followed Amr Khaled's footsteps and emulated his approach by adopting the image of a young and modern non-formally trained preacher, who is successful in communicating with youth through social media platforms.

It is worth noting that the image of some of these religious online influencers was shaken and their credibility was somewhat diminished in the aftermath of the Arab Spring uprisings, due to their tendency to avoid delving into thorny and sensitive political issues which question the authority of dictatorial regimes in the Arab region.

On a global scale, one good example is Yasmine Mogahed, a young Egyptian-American woman preacher who shares the same qualities as Khaled and Hosni. However, she appeals to a diverse international audience, on a global scale, rather than an Arab audience only, since her online lessons and speeches are all in English, unlike Khaled and Hosni, who address their audiences in the Arabic language. This highlights the importance of the language of communication in determining the outreach, visibility, and impact of different religious scholars in the digital age.

Interestingly, while there is usually a demarcation between the older and more formally trained Imams, who mostly lack digital literacy skills, and the younger more modern Imams, who are mostly more technologically savvy but may lack the needed religious knowledge and formal education, there are some exceptions to this rule. Occupying an in-between third position between the first and second groups, there are the "hybrid Imams" (Patel 2022), who combine both offline and online tools and techniques. Many of them are also classified as "celebrity Imams" (Patel 2022), such as Yasir Qadhi, Suhaib Webb, Mufti Menk, and Omar Suleiman, all of whom have a strong presence, high visibility, and a huge following online, in addition to a sound reputation offline based on authoritative religious knowledge, simultaneously.

Another good example of a hybrid Imam who was able to create a successful crossover between tradition and modernity was the late Egyptian, Qatar-based cleric, Shaikh Youssef Al-Qaradawi, who became very popular thanks to his regular appearance on a television show on *Al Jazeera*, as well as his establishment of the equally popular website *Islam Online* (Bunt 2018, p. 73).

Adopting the position of the school of *wasatiya* (middle ground of moderation), which by definition advocates for a balanced, middle position, as the Arabic term implies, Qaradawi helped launch the pioneering website *Islam Online*, which became very visible, popular, and influential globally, thanks to its wide international outreach, multilingual postings, attractive page design, and, most importantly, the credibility of its authenticated religious information (El-Nawawy and Khamis 2009).

With their hybrid modes of transmitting religious knowledge, coupled with their highly esteemed celebrity status as credible religious figures in the Muslim community who combine formal Islamic education with a strong social media presence and a huge online following, the hybrid celebrity Imams play an important role in resolving many young Muslims' dilemmas when it comes to the authority and authenticity of Islamic knowledge. This is especially the case for new generations of Muslims living in the West because "[w]hen navigating through the limitless Islamic resources and digital *fatawa* on the internet, young Muslims. . .not only have to choose between different imams for guidance, they also have to define what is 'authentically' Islamic" (Patel 2022, p. 49).

However, the religious information overload on the internet, due to the massive volume of available online fatwas, websites, profiles, Facebook pages, and Twitter accounts—not to mention the increasingly popular Instagram and TikTok accounts—had a counter effect on the credibility of religious knowledge, its authority, and authenticity, even when the source is a hybrid celebrity Imam.

In fact, "[w]hile it is true that having unlimited access to religious knowledge on the internet is beneficial for many young Muslims, the unprecedented volume of information also presents doubts and tensions" (Patel 2022, p. 49). This resulted in an increasing phenomenon whereby many young Muslims today, especially those in the West, "displayed a hesitation to default to one or more resources in relation to Islamic questions. They always questioned the legitimacy of sources of information that they found online . . . Similarly, they also questioned the authenticity of the celebrity *imams* that they themselves 'followed' or 'liked' on social media" (Patel 2022, p. 49).

What is clearly witnessed here is a cacophony of different voices representing varying degrees of authentic Islamic knowledge, or lack thereof, who are constantly competing for their audiences' attention, approval, and following, in an effort to create or maintain their status and credibility in an increasingly crowded digital space, where central religious

authority and authenticity seem to suffer. This, in turn, invites the danger of spreading religious misinformation and/or disinformation which could widely proliferate under the mantle of promoting religious reform or modernizing traditional religious beliefs and practices in the Muslim world.

It is important to bear in mind that such phenomena expanded globally to encompass other geographic regions, such as Southeast Asia, especially in Muslim-majority countries, like Indonesia, for example. In his investigation of "how the practice of mediatization shapes and reconfigures Muslim religious authority and the parameters of Islamic authenticity" (Alatas 2022, p. 51), Ismail Fajrie Alatas "examines the engagement of a contemporary Indonesian Sufi master, Habib Luthfi Bin Yahya (b. 1947), with different media forms to explore how religious authority takes shape and is configured through different practices of mediatization" (Alatas 2022, p. 51).

His analysis of this case study revealed how this Sufi master was able to successfully become a "visual master" (Alatas 2022, p. 59) through skillfully deploying a wide range of social media tools and techniques, including livestreaming his large monthly gatherings with his disciples on Facebook, creating a prominent online platform for his popular magazine, and maintaining an active presence on his Facebook page, Twitter account, and other social media platforms. His technological savviness, coupled with his spiritual authority, helped grow his fan base, boost his credibility, and increase his visibility significantly as a Sufi leader and a religious scholar (Alatas 2022).

Likewise, it is important to take note of how these new modes of reinvented Islamic religious authority in cyberspace are not just confined to Sunni Muslim countries or communities. Rather, they are also prevalent among Shiite communities. For example, in his study of how "Shi'i clerical authority has undergone a performative network transformation with digital technology playing a key role in the process" (Rahimi 2022, p. 222), Babak Rahimi provides several examples of how contemporary Shi'i religious scholars of different ranks "engage in diverse modes of authorial performances by carving out new spaces of public presence—an alternative landscape where self-representation becomes, despite physical absence, accessible, spreadable, and inherently more visible" (Rahimi 2022, p. 222).

These complex phenomena are illustrated through providing varied examples of how Shi'i scholars from different age groups and ranks are relying on social media platforms, including their Instagram accounts, to reach broader audiences, both inside and outside Iran, and how they are enhancing their spiritual and moral authority, through presenting themselves as ordinary people engaging in the most mundane everyday acts, such as riding a bike or celebrating a child's birthday, rather than just engaging in formal religious duties such as giving Friday *Khutbas* (religious sermons), for example (Rahimi 2022).

The existence of these new models of modern Islamic religious authority undoubtedly opened the door for many new transformations and dynamics, including new forms of *ijtihad* (interpretation of Islamic texts and teachings), new models of religious authority and scholarly leadership, and new identity markers and manifestations in the digital age (El-Nawawy and Khamis 2009).

One important internet-related paradox impacting contemporary Muslim communities is the tension between hegemony and resistance, as well as tradition and modernity, in the religious sphere. It is important to bear in mind that, on one hand, the internet provides platforms for traditional Islamic scholars and traditional institutions of Islamic knowledge, such as *Al Azhar* in Egypt, for example, while, on the other hand, it also provides an opportunity for more modern, innovative, or even controversial voices to be heard.

Moreover, it is important to bear in mind that these newly created hybrid digitalized spaces have the potential to increase the numbers of those who can contribute to "creating and sustaining a religious-civil public sphere" (Eickelman and Anderson 2003, p. 14) which falls outside the realm of mainstream, authoritative, and traditional sources of Islamic knowledge.

Given the survival and continuation of many forms of Islamic tradition, it is important to remember that the internet, and its many related applications, could provide its users

with the needed tools and potentials to question, critique, redefine, and reinvent many of the long-held Islamic traditions and practices. That is certainly true because "[a]ccess to canonical religious resources—the *Qur'an*, *hadith* (traditions of the Prophet Muhammad), and the sources of Islamic jurisprudence (*fiqh*) is no longer the exclusive monopoly of religious scholars and the educated elite" (Rozehnal 2022, p. 6).

This new phenomenon is best exemplified in the "self-proclaimed religious authorities" who appear online and have their own followers and critics (Bunt 2018, p. 81). These authorities provide religious judgment and advice through online-based platforms, such as "fatwa-online, Ask-Imam, and Islam Q&A" (Bunt 2018, p. 81).

In addition to the realm of Islamic religious authority, it is equally important to investigate the paradoxes of modern Islamic discourses when it comes to shaping and reflecting the evolving gendered identities of Muslim women in the age of cyberspace and the controversies associated with this phenomenon.

### 3. The Paradoxes of Gendered Islamic Identities in the Digital Age

Beside the paradoxes taking place in the realm of religious authority, there is an equally significant transformation taking place in a parallel domain, namely the social domain, which is worth careful investigation. This is especially important when addressing Muslim women's gendered identities in cyberspace and their varied manifestations and expressions.

The internet, and its many related applications, provide Muslim women with unprecedented opportunities to exercise their agency, amplify their messages, and widen their outreach in the political, social, cultural, and religious domains simultaneously (Khamis 2022).

Here again, just like the case of Islamic religious authority online, Muslim women's mediated activities and online identities do not reflect a "one size fits all" position. Rather, they represent a wide array of identity markers which constantly oscillate between tradition and modernity in myriad ways and through various forms of expression.

While some Muslim women chose to utilize the internet to exercise and express their religiosity online and build faith-based nurturing communities of support and solidarity, others chose to use the internet to express unorthodox controversial identities which are meant to bridge the gap between tradition and modernity in unconventional ways, triggering mixed reactions.

One good example exemplifying the first category is the Arabic and English "Islam Way Sisters Discussion Forums" which provided a uniquely exclusive Muslim women-only safe space for the discussion of religious, social, and personal issues. This reflected a form of "Islamic feminism" (Khamis 2010) and "Islamic sisterhood" centered around faith and gender (El-Nawawy and Khamis 2009).

Other examples of this category of traditional and religiously conservative Muslim identities online include Muslim women-only closed Facebook groups such as "Monthly Sisters' *Halaqa*" (religious study circle), which boosts the participants' spirituality and provides them with the necessary religious knowledge, through studying, reciting, and memorizing the holy Qur'an and sharing useful religious sermons (Khamis 2022).

Another Muslim women-only closed Facebook page is "Surviving *Hijab*" (Islamic headscarf). Its declared aim is to provide the necessary moral and emotional support to those who need it the most, such as Muslim women who are thinking about wearing the hijab, those who are thinking about taking it off, and those who took it off but are considering wearing it again (Khamis 2022).

On the other hand, good examples of the second category of modern and controversial identities online include the new phenomenon of so-called *hijabista*, referring to "a 'Muslim woman who dresses 'stylishly' while still adhering to an array of 'modest' apparel that coincides with Islamic dress code" (Waninger 2015, p. 2). Modeling after the concept of "fashionista," which refers to a fashion icon and role model, especially in the digital sphere, the new concept of *hijabista* combines the terms *hijabi* (a woman wearing the Islamic headscarf) and fashionista. These new young Muslim women personas, who combine modesty with fashion in the online sphere, have become very popular among

young Muslim women in recent years, setting a new digital trend of Islamic social media influencers (Khamis 2022).

They play the role of "influencers" and role models who could be best defined in this context as the "fashion public opinion leaders" and "fashion agenda-setters" (Khamis 2022, p. 105), who play a significant role in setting an example for other Muslim women, especially young women, who are keen to combine modesty, elegance, and style (Khamis 2022, p. 105).

This new phenomenon of online *hijabistas* is not free of tensions and controversies, however, despite its wide appeal among young women and its glamor (Khamis 2022, p. 106). This is mainly because "the digital realm allows for opportunities for multiple constructions of self" some of which "revealed both an Islamic religio-cultural identity and a fashionable Western identity, at times emphasizing one more than the other, at times combining the two in unorthodox ways" (Kavakci and Kraeplin 2017, p. 856).

This unconventional and hybrid phenomenon triggered equally hybrid and mixed reactions from diverse Muslim communities online, depending on gender, generation, religious orientation, and geographic location, among other factors, with the younger and more cosmopolitan internet users in the diaspora mostly expressing admiration or even fascination, while the older, more conservative, religiously orthodox internet users from within the Muslim world mostly criticizing these new stylish fashion icons and their Western-influenced fashionable styles as the antithesis of modesty and religiosity (Kavakci and Kraeplin 2017). In every case, these new negotiated identities are reflective of "the experience of duality or multiplicity that confronts hijabis with an active online presence" (Kavakci and Kraeplin 2017, p. 866).

Needless to say, the phenomenon of *hijabistas* is just one example among many when it comes to expressing modern, unconventional, or even controversial Muslim gendered identities online. Other examples include young Muslim women who decided to engage in various forms of artistic expression, including hip hop and rap music, to reflect their mediated hybrid positionalities, shifting narratives, and eclectic worldviews.

There are also groups of Muslim women activists that were able to take advantage of cyberspace to amplify their voices and defend their causes. One example is the group "Women Living Under Muslim Laws" (WLUMS), which is a global organization that draws attention to women's conditions across different Muslim countries. It disseminates messages online in seven different languages to reach the largest possible global audience, thus amplifying the voices of culturally diverse Muslim women online (Bunt 2018, p. 76).

By doing so, it could be said that this group is engaging in a digital form of "Islamic feminism" (Khamis 2010) which takes advantage of new media tools and techniques to advance Muslim women's rights and to raise more awareness about their grievances and any injustices they may face in their societies, by sharing these controversial issues with a diverse global audience worldwide in multiple languages and across different platforms.

The nature of the controversial and sensitive topics which this group tackles triggers a division of opinion. On one hand, the fans and supporters of this group admire and praise its actions and perceive them as necessary efforts to bring about much-needed change, and possibly reform, in contemporary Muslim societies when it comes to women's issues, especially issues of gender equity in the political, social, cultural, and legal spheres. On the other hand, however, this group's critics are skeptical of some of the issues it defends, some of which remain controversial in Muslim societies, such as LGBTQ rights in Muslim communities, criminalizing marital rape, and supporting the anti-*hijab* movement in Iran, to mention a few examples.

Such controversies, tensions, and debates are indicative of the push-and-pull mechanisms and heated controversies, not just around controversial topics, but, rather, around the very concept of Muslim identity itself, and its myriad definitions, boundaries, and determinants.

Another unique illustration of Muslim women's expression of their faith and identity in cyberspace (Piela 2012) was the creation of the so-called "digital *Niqabosphere*" (Piela 2022) as a hypermediated third space for women who wear the *niqab* (face veil) to ex-

ercise their agency and express their identities online (Piela 2022). "The *digital niqaboshpere* offers a rare chance to interact with a *niqabi* to many people who otherwise would not have this opportunity. For the women themselves, these settings facilitate mobilization around shared values, beliefs, and experiences" (Piela 2022, p. 144).

The picture which emerges from this overview of various forms of gendered expressions and positionalities by different groups of Muslim women representing different generations, worldviews, backgrounds, and varying degrees of religiosity online is reflective of the nuanced, complex, and hybrid concept of "Islamic feminism", which is an equally paradoxical phenomenon in many ways (Khamis 2010).

These tensions and paradoxes also illustrate how various "counter publics" (Cesari 2004), who may be marginalized on the basis of different traits including gender, generation, sect, or theological school of thought, could create communities of solidarity and support online which provide safe spaces for everyone to express their views, negotiate their identities, and stand up with and for each other.

## 4. Concluding Remarks

This critical essay illustrates how, in the middle ground between the opposing push and pull forces of tradition and modernity, uniformity and divergence, hegemony and resistance, and localization and globalization and their associated new manifestations in cyberspace, new hybrid, negotiated, eclectic, and unconventional identities are born in the age of digital transformation, in both the religious and social spheres simultaneously.

It also illustrates how the new internet-based communication platforms and channels played an effective dual role in both shaping and reflecting the hybridity, complexity, and dynamism of these evolving identities at the individual and collective levels, across various spatial and temporal contexts.

This reminds us that the transnational borderless nature of the internet has mutually enabled the creation of both joint platforms for the display of collective identities within the realm of the 'virtual *umma*' (Islamic community) among Muslims in the digital age, as well as providing forums for divergent identities to freely express themselves. Thus, the internet plays both a divisive and an integrative role simultaneously between different Muslim identities (El-Nawawy and Khamis 2009).

In other words, the internet and its related applications and usages, such as blogs and discussion boards, for example, fuel the dual, yet contradictory, functions of aiding uniformity and divergence simultaneously. On one hand, there is evidence of increasing uniformity, solidarity, and cohesion between Muslims in cyberspace who share a "collective identity" as members of the same *umma*. On the other hand, however, there is evidence of increasing divergence, through widening the gap between "divergent identities" which are demarcated on the basis of different demographic, geographic, and ideological factors (El-Nawawy and Khamis 2009).

In examining these complex and hybrid Muslim online identities, special attention must be paid to how the participation in the realm of cyberspace helps promote a sense of religious communalism and collectivism, which allows members of the Muslim *umma* to (re)construct their identities as members of the same community of faith, while also discovering the differences among themselves and demarcating themselves from non-Islamic practices and lifestyles, on one hand, and variations in Islamic faith and practice, on the other hand.

The ongoing process of redefining and renegotiating local cultural and religious identities in a global context, i.e., "globalizing the local" (Mandaville 2001, p. 76), could result in three potential outcomes for Muslims today, especially Muslim youth. It can unify and integrate them under the banner of globalism; it can separate them by making them more aware of their own internal differences and magnifying their sense of "otherness;" or it can create an 'in-between' status, where a 'hybridized identity' embraces elements of both the diaspora and the homeland simultaneously (Mandaville 2001).

The discussions in this critical essay support the third negotiated middle position, since they speak to the coexistence of parallel, yet paradoxical, trends in the realms of Islamic religious authority and shifting Islamic gendered identities with numerous manifestations, complexities, nuances, and hybridity in the realm of cyberspace and beyond.

It is safe to conclude that the push-and-pull mechanisms and tensions between these paradoxical and contrasting forces will continue to grow and expand both online and offline in Muslim societies moving forward, creating new middle positions between the poles of the traditional and the modern, the old and the new, the local and the global, and the religious and the secular in a cyclical ongoing phenomenon which reflects the equally shifting, transitioning, and paradoxical realities of contemporary Muslim societies.

**Funding:** This research received no external funding.

**Institutional Review Board Statement:** Not applicable.

**Informed Consent Statement:** Not applicable.

**Data Availability Statement:** Data are contained within the article.

**Conflicts of Interest:** The author declares no conflicts of interest.

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
