# Peer review of "The Paradoxes of Modern Islamic Discourses and Socio-Religious Transformation in the Digital Age"

_religions, doi:10.3390/rel15020207_

Round 1
Reviewer 1 Report
Comments and Suggestions for Authors
The article can be improved by explaining the methods and design of the research. There should be a section on research methodology explaining the methodology and why this particular methodology is the best approach for this paper. There should be a section on the literature review: the author needs to tell the readers about past research on this topic and then say a few words about how this research builds on and departs from past research. What makes this research different, unique, and original? We need further discussion/analysis on the takeaways: What do these findings mean for Muslims and Islam? Why is this research important? What is the goal? What does the author want to accomplish from this research? Further discussion and explanation would be helpful.
Author Response
-Regarding the comment in Report 1 about adding a methodology section which details the research questions, hypotheses, objectives, etc., this article is a critical essay which discusses the topic at hand and attempts to unpack it, analyze it, and better understand it, but it is not based on collecting empirical data. This type of article does not require a methods section. I just added in the introduction that this is a critical essay to make this distinction clear.
-As for the comment in Report 1 about adding a literature review section, the appropriate review of relevant previous studies in addition to current studies is included in the introduction section as well as throughout the entire article. It is not required to have a separate literature review section in this type of critical essay.
-As for the comment in Report 1 about discussing the takeaways from this study and why it matters to Islam and Muslims, this is thoroughly discussed throughout the article and also in the concluding remarks section at the end.
Reviewer 2 Report
Comments and Suggestions for Authors
The research design, questions, hypotheses and methods are absent.
Information regarding the research question and research method are important elements of academic writing.
Author Response
-Regarding the comment in Report 2 about adding a methodology section which details the research questions, hypotheses, objectives, etc., this article is a critical essay which discusses the topic at hand and attempts to unpack it, analyze it, and better understand it, but it is not based on collecting empirical data. This type of article does not require a methods section. I just added in the introduction that this is a critical essay to make this distinction clear.
Reviewer 3 Report
Comments and Suggestions for Authors The subject of the research is very interesting. However, the study does not provide any new information. The paper seems more like a summary of some previous studies.
Author Response
-I beg to disagree with the comment made in Report 3 that this article doesn’t add something new. A lot of the discussions in this article, especially those related to new and hybrid Muslim identities in cyberspace, such as the hijabistas phenomenon, for example, are new and so are the sources tackling them which are as recent as 2021 and 2022.
Reviewer 4 Report
Comments and Suggestions for Authors
Some of the information and references are outdated; you need to update your argument. It is true that many of the Muslim scholars you cited (mainly Amr Khaled and Mostafa Hosni) were very influential in the early 2000s but after the Arab Spring Amr Khaled and Mostafa Hosni, among others, lost the appeal they had earlier and many their fans have forsaken them because, among other reasons, they have hardly debated the idea of dictatorship prevailing in the Arab and Muslim or the injustices from which millions of Muslims suffer throughout the Arab and Muslim world. Amr Khaled and Mostafa Hosni and their likes would never touch on matter that may anger unjust Arab and Muslim rulers; they only preach about worshiping practices (how to make wudu’ or how to take a shower after making love) and such things. To make your paper stronger, you may need to examine how the reception of the scholars you cited changed (favourably or unfavourably) before and after the Arab Spring and after.
Author Response
-I appreciate the valid point made in Report 4 about the impact of the Arab Spring on the credibility of some of the religious social media influencers, such as Amr Khaled and Moustafa Hosni. I added a few sentences explaining this point and addressing this legitimate concern regarding their challenged credibility in the post-Arab Spring era, although this is not the main focus of this article.
Round 2
Reviewer 1 Report
Comments and Suggestions for Authors
The essay is a timely contribution to the field of Islamic studies. Further research will emerge from this piece. This project meets the requirement for a critical essay but not an academic article, for the latter requires further academic research, sound research methodology, and further analysis and implications. Since the author replaced the term article with critical essay in the abstract, this should be the case throughout the essay.
Author Response
I would like to thank Reviewer #1 for accepting my essay for publication and for the good and kind feedback he provided on it.

Reviewer 3 Report
Comments and Suggestions for Authors
In the improved version, I hardly see any improvements. My points remain valid. See my comments in the pdf.

Author Response
I hope that Reviewer #3 would appreciate the fact that this is not an empirically-based academic study and, therefore, the previously provided feedback about including a hypothesis or hypotheses and a methodology section, etc. is not applicable in this case.
